psychology

autism, communication, language, theory of mind, pragmatics

**Author for correspondence:**
A. C. Wilson
e-mail; alexander.wilson2@psy.ox.ac.uk

# Judging meaning: A domain-level difference between autistic and non-autistic adults

## A. C. Wilson and D. V. M. Bishop

Department of Experimental Psychology, Oxford University, Oxford, UK

ACW, 0000-0001-7077-1618

We tested whether autistic adults would show selective difficulties across several tests of inferencing and social understanding in the context of average-range core language ability. One-hundred and ninety-one participants completed an online battery, and data were analysed using confirmatory factor analysis. Results showed that vocabulary knowledge was separate from other measures, which collectively formed a 'receptive communication' factor. Autistic people underperformed on the 'receptive communication' factor but showed more advanced vocabulary knowledge than non-autistic participants. Lower performance among autistic adults on the test battery predicted face-to-face communication difficulties measured by self-report and researcher ratings, with moderate effect sizes. Follow-up analysis indicated three further findings. We hypothesized that differences would arise from an isolated 'theory of mind' difficulty in autistic people, but instead the data suggested more general information-processing differences when making judgements about communicative stimuli. Second, substantial group differences on a test of implied meaning were only partly explained at the factor level, suggesting that multiple cognitive influences underpinned these differences. Finally, autistic women tended to perform better than autistic men. Our results support the idea of a subtle domain-level difference in pragmatics in autistic people, while questioning the basis of this difference and highlighting substantial variability in skills across groups.

## 1. Introduction

Difficulties with back-and-forth conversation are a diagnostic feature of autism [1]. As many autistic people do not have impairments in core aspects of language such as grammar and vocabulary, these conversational difficulties are typically attributed to pragmatics (e.g. [2,3]). Pragmatics underpins (i) the functional and

communicative *use of language*, such as the selection and maintenance of conversation topics, as well as (ii) a person's *understanding of language* in context, for instance reading between the lines of what someone is saying. Both aspects are likely to be a challenge for autistic people in day-to-day communication, but for the purpose of this work, we focus on pragmatic understanding—that is, inferring the full extent of a speaker's meaning in the communication context. Pragmatics is a language domain that has traditionally been challenging to assess, especially because existing tests do not convincingly separate pragmatics from other aspects of language function. Responding to this limitation, we developed a novel Implicature Comprehension Test, which requires the test-taker to process implied meanings in conversations that are closely controlled to reduce other linguistic demands. Pragmatics could be disentangled from core language skills using this test [4]. We reported on a factor analysis including this and several other novel tests administered to a non-autistic sample and found that the Implicature Comprehension Test clustered with other tests requiring social understanding, in a two-factor model with 'core language' and 'social understanding' as the two factors.

Here, we extend the factor analysis to include autistic adults alongside the non-autistic sample, in an analysis pre-registered with the open science framework (OSF); see below for details on the pre-registration. We had three hypotheses:

(1) The tasks would dissociate into two factors—'social understanding' and 'core language'—rather than all grouping under one general factor.
(2) Autistic adults would show differences in pragmatic understanding, reflected in lower scores on the 'social understanding' factor, but no differences in 'core language' ability.
(3) Variability in researcher-rated and self-rated communication skills among the autistic sample would relate to scores on the 'social understanding' factor.

The first hypothesis involved a replication in autistic adults of the factor structure observed in non-autistic adults. In the second hypothesis, we aimed to account for possible differences in pragmatic understanding on the Implicature Comprehension Test through a domain-level difference in social understanding. The 'social understanding' factor might be seen as overlapping with 'theory of mind', and so this hypothesis is in line with the idea that the social communication challenges of autistic individuals emerge through differences in the processes by which we keep track of and understand other people's mental states [5]. Given the view of relevance theory that pragmatic understanding is embedded within our broader abilities to understand others, i.e. our 'theory of mind' capacity [6], it is plausible that differences in performance on specific tests with a pragmatic demand should be accounted for by overall differences at the level of this 'social understanding' factor. In terms of empirical support for the 'theory of mind' account of autism, reviews have extensively documented differences in the performance of autistic groups compared with non-autistic people on 'theory of mind' tasks, involving false beliefs and other complex mental states [7,8], although it should be noted that there are issues surrounding construct validity and failures to replicate that have led to some dissatisfaction with the 'theory of mind' account (e.g. [9]). In addition, we note that definitions of 'theory of mind' are not consistently agreed-upon and 'theory of mind' tests often tap related constructs such as emotion perception, figurative language comprehension and empathy; with the exception of figurative language, meta-analyses provide support for average differences between autistic and non-autistic people in some aspects of these other constructs (see [8,10,11]). In addition to a difference on the 'social understanding' factor, we hypothesized that there would be no difference on 'core language' factor between autistic and non-autistic groups, as structural language impairments are not a core feature of autism [1]. However, it should be noted that language outcomes are highly heterogeneous in autistic adults [12]. The final hypothesis focused on variability in autistic adults: we expected global measures of social communication (based on researcher-rated face-to-face interaction and self-report) to inter-correlate with scores on the 'social understanding' factor, thereby indexing a single communication continuum along which people would vary.

## 2. Methods

Analysis reported here was pre-registered on OSF (https://osf.io/3ecr4). Note that this pre-registration included several analyses, one of which focused on mental health and is reported elsewhere [13].

Ethical review for this study occurred in two stages, separately for the autistic and non-autistic participants, as the groups were recruited sequentially. The first stage of the study was granted ethical clearance in July 2018 (Ref. R57087/RE002) and the second in November 2018 (Ref. R59912/RE002) by the Medical Sciences Division of the Oxford University Research Ethics Committee.

## 2.1. Power calculation

The sample size rationale was based on power analysis. We simulated 1000 datasets ($N = 170$) from a population model with a two-factor correlated traits structure. Two variables were set to load on a 'core language' factor and four variables on a 'social understanding' factor; factor loadings were 0.6 and 0.8 for the 'core language' factor, and 0.7, 0.5, 0.4 and 0.4 for the 'social understanding', based on the factor structure observed in the control participants. The two factors were set to correlate at 0.5. Autism diagnosis was included in the model as a dichotomous covariate and set to correlate with the 'social understanding' factor at 0.3. All simulated datasets were fitted to two models, the two-factor model and a one-factor model in which all variables were set to load on a general factor. Model fit was compared using a chi-square test. A significant effect ($p < 0.01$) was found in all simulations (power for hypothesis 1). We then fitted each dataset to two-factor models which included diagnosis as a covariate: in one model, the 'social understanding' factor was allowed to covary with diagnosis, and in the comparison model, it was not. Fit was compared between these two models using a chi-square test. A significant effect was found in all datasets (power for hypothesis 2).

## 2.2. Participants

We recruited autistic adults through support and social groups, and through Autistica, a research-focused charity in the UK. Inclusion criteria for individuals giving informed consent to participate included: (i) an autism spectrum diagnosis by a clinical service, (ii) native-level fluency in English, and (iii) age of 18 years or over. Exclusion criteria included: (i) significant visual or hearing impairment, (ii) history of neurological illness or head injury, and (iii) no access to a computer with Internet access and audio. Individuals were invited to participate regardless of other diagnoses, including ADHD, genetic syndromes or learning disabilities. We aimed to recruit at least 50 autistic individuals, in line with the power calculation, but rather than prescribe a set sample size, we set a specific date as the stopping rule for recruitment: the study opened in 2019 and all individuals expressing interest in participating by 31 March 2019 were invited to do so.

The comparison sample of 120 non-autistic adults was recruited online via the participant platform, Prolific (https://www.prolific.co/). They fulfilled similar inclusion criteria, with the exception of having no autism diagnosis. This is the same sample as recruited in Wilson & Bishop [4]. Average age of the non-autistic participants was 30 years, 11 months (s.d. = 11 years, 3 months; min = 18 years, max = 64 years). Sixty-five identified as women, 54 as men and one person did not declare their gender. The majority of the sample described themselves as White (103 out of 120); four people identified as Mixed Race, four as Black and eight as Asian. Thirty-four people said they were currently students. Of the 86 individuals who reported not being students, the highest level of education was given as high school/secondary school for 18 individuals, vocational training/college courses for 13, bachelor's degree for 53 and a higher degree for nine.

The autistic sample comprised 71 people. Of 80 individuals recruited into the study, 74 completed the battery of cognitive tasks which forms the basis of the analysis here, and of these 74 people, 71 reported a clinical diagnosis by appropriately trained professionals (clinical psychologists, psychiatrists and specialist nurse practitioners trained in autism diagnosis) and mostly as part of multidisciplinary teams (MDTs) in National Health Service (NHS) settings. Forty-five individuals identified as female, 25 as male and one as non-binary. Average age was 38 years (s.d. = 14 years, min = 18 years, max = 70 years). The approximate average age at diagnosis was 31 years (s.d. = 18 years, 50 individuals were diagnosed as adults). It should be noted that the gender distribution of the sample was rather different to population norms for diagnosis (as more males receive a diagnosis) and a particularly high proportion of individuals were diagnosed in adulthood. As part of the research protocol, the majority of participants ($n = 65$) took part in an Autism Diagnostic Observation Schedule Second Edition (ADOS-2) assessment as noted below; six participants did not for logistical reasons. Fifty-five people scored at or above the ADOS-2 'autism spectrum' cut-off. Of the ten who did not, all were female, which speaks to the notion that autistic females may mask some of their difficulties, especially in 'safe' one-to-one situations such as during an ADOS-2 evaluation [14]. Other neurodevelopmental diagnoses reported by participants included: dyslexia ($n = 8$), language disorder ($n = 3$), dyspraxia/developmental coordination disorder ($n = 9$) and ADHD ($n = 7$). Except for one Asian person, each autistic participant was White. Fifteen people said they were currently students. Of the 56 individuals who reported not being students, the highest level of education was given as at least some high

school/secondary school for 8 individuals, vocational training/college courses for 16, bachelor's degree for 16 and a higher degree for 15 (one person did not answer this question).

## 2.3. Procedure

For autistic participants, the study involved three sections, the first two of which took place online. In the first section, individuals were asked to provide information about their diagnosis and completed three self-report questionnaire measures including the Communication Checklist – Self Report (CC-SR; [15]). In the second section, individuals completed a set of seven cognitive tests online at a time and place of their choosing. The tasks were supported by Gorilla, the online tool for behavioural experiments (https://gorilla.sc/). In the third section, participants were either invited to the Department of Experimental Psychology, University of Oxford, or visited at home, for an in-person communication assessment using Module 4 of the ADOS-2 [16], and completed further cognitive tasks under supervision. Individuals were compensated with a £20 voucher for their participation.

Non-autistic participants only completed the second section of the study, detailed above. They were compensated £5.

## 2.4. Measures

We briefly outline measures used in the factor analysis, which were given in the second section of the study protocol and are fully detailed in Wilson and Bishop [4]. Then, we describe the global measures of social communication skills and further cognitive tasks used in the analysis.

### 2.4.1. Tests expected to tap 'social understanding'

#### 2.4.1.1. Implicature comprehension test
Participants watched short cartoon videos of conversations, in which two characters each produced a short utterance, followed by a comprehension question eliciting a response of 'yes', 'no' or 'don't know'. For 36 items, participants needed to process an implicature to answer the question; note that in the data collected from the non-autistic adults, one of these items was poorly functioning, so was excluded from the test. There were also 10 items where the answer was more explicit; these served as positive control items. The main measured variable was the sum of implicature items correctly answered (out of 36), and there was also a control variable, the sum of explicit-response items correctly answered (out of 10).

#### 2.4.1.2. Test of fillers and backchannels
Participants watched videos in which a character made a short utterance; a second character then produced a word before the audio cut-off. Participants needed to select which of the two characters was likely to be speaking now; they could indicate that it was 'very difficult to say' as a third option. The task included 40 items. For half the items, the second character produced a backchannel continuer (mm-hmm, uh-huh, really, right, huh?) and for the other half, a filler claiming the floor (um, uh, yeah, oh, well). Where a backchannel continuer was used, the first character was assumed to be the person likely to speak, and the second character where a filler claiming the floor was used. There was one measured variable: the sum of items correctly answered (out of 40).

#### 2.4.1.3. Awkward dialogues
Participants listened to eight short dialogues with two characters each taking five turns. In five dialogues, a character said something socially awkward. In three, nothing awkward was said. Participants indicated if something awkward was said and typed a couple of sentences to explain what was awkward, why it was said and how the characters might have felt. For each of the five awkward dialogues, participants were given a score out of two by two independent markers who showed reasonable agreement (Cohen's kappa = 0.73), and discrepancies were resolved through discussion. After each dialogue, participants were asked a factual recall question. If the participant did not score two out of two for an awkward dialogue and did not answer the factual recall question correctly, that item was recorded as invalid, since the participant was unlikely to have sufficiently processed the basic details of the scenario, so would not be well-placed to judge it as awkward or not. If more than one item was invalid, the whole test was recorded as invalid. Raw scores on the five awkward dialogues were processed into factor scores using item response modelling as the measured variable for this test.

### 2.4.1.4. Frith-Happé animations [17]

In this test of mental state attribution, participants watched a sequence of animations showing two moving triangles that interact. Two item types were presented during this study: 'theory of mind' ($n = 5$) and 'goal-directed' animations ($n = 4$). In the 'theory of mind' animations, the triangles interacted as if they were trying to influence the thoughts or feelings of each other, whereas in the control 'goal-directed' animations, they interacted physically (e.g. by chasing each other). Participants provided a typed description of what they thought happened in the animation. Each of these descriptions was scored out of three according to the original mark scheme by two independent markers who showed reasonable agreement (Cohen's kappa = 0.70), and discrepancies were resolved through discussion. This gave nine separate scores: for the five 'theory of mind' animations and the four 'goal-directed' ones. The main measured variable was a factor score for the 'theory of mind' animations, processed using item response modelling. We also processed total scores for the control animations for use in the follow-up analysis.

### 2.4.2. Tests expected to tap 'core language'

### 2.4.2.1. Synonyms test (receptive vocabulary)

This included 25 items in which participants were presented with words on the screen and chose which of four words was a synonym for the target word. The task was timed (up to 12 s per item). There was one measured variable: the sum of items correctly answered (out of 25).

### 2.4.2.2. Grammaticality decision test (receptive grammar)

Participants listened to sentences and decided if they were well-formed and grammatical; there was a 6 s limit to listen and respond. There were 44 items (although 50 items were included in the original version of the test, six were poorly functioning, so have been dropped). There was one measured variable: the sum of items currently answered (out of 44).

### 2.4.3. Global communication measures

### 2.4.3.1. Communication checklist – self report [15]

This questionnaire consists of 70 statements (50 reflecting communication difficulties and 20 reflecting communication strengths), for which participants rate on a four-point Likert scale how frequently each applies to them; the scale is from 'less than once a week (or never)' to 'several times a day (or all the time)'. As we were interested in pragmatic communication difficulties, we extracted the pragmatic language composite from the checklist. An example item is 'People tell me that I ask the same question over and over' (pragmatic language). Where participants left two or fewer items of the pragmatic language scale unanswered, the total score was prorated; if more than two items were unanswered, the total score was recorded as missing for that participant.

### 2.4.3.2. Module 4 of the autism diagnostic observation schedule version two ([16])

This involves several structured tasks, free conversation and interview-style sections, which the participant carries out with a trained researcher. Individuals are rated on a range of behaviours, and scores on an 11-item subset allow classification of individuals as 'non spectrum', 'autism spectrum' or 'autism'. Scores on these 11 items are totalled into a communication and social interaction index; for simplicity, we refer to ADOS-2 total.

### 2.4.4. Further cognitive variables

### 2.4.4.1. Test of local textual inference

This measure of narrative-based inferencing was administered during the second section of the research protocol to all participants. In our original study in which these tasks were administered to non-autistic individuals [4], we expected this test to load on the 'core language' factor, but it actually seemed more

closely linked to 'social understanding'. As this was unexpected, we did not include it in our pre-registered analysis for this study, but reserved it for follow-up analysis. In the task, participants read two brief sections of a short story and after each section they responded to 10 questions with a word or short phrase. Participants needed to make inferences based on the short story to answer the questions. The text remained on the screen as the questions were asked. There was one measured variable: the sum of items correctly answered (out of 40).

### 2.4.4.2. International cognitive ability resource sample test [18]

This test of general ability was administered in the third section of the research protocol, only to autistic participants. There were four items of four different types: matrix reasoning, verbal reasoning, three-dimensional rotation and letter-number sequences. The International Cognitive Ability Resource (ICAR) sample test has been validated in a large online sample, has good internal consistency (alpha = 0.81) and correlated with the Shipley-2, a commercial IQ measure, at 0.81 when corrected for reliability and restriction of range. Data for the ICAR sample described above is freely accessible on Dataverse (http://dx.doi.org/10.7910/DVN/AD9RVY). As young college students are significantly over-represented in the dataset, population norms cannot be adequately generated from it, though we used the data as a rough comparison for the distribution of scores in the autistic adults involved in this study. There was one measured variable: the sum of items correctly answered (out of 16).

## 2.5. Data analysis

Data were analysed in R [19]. Data and scripts are accessible on OSF (https://osf.io/b9t8a/). R package SemPlot [20] was used to make factor model diagrams and yarrr [21] to make the pirate plot.

### 2.5.1. Preliminary analysis

In Wilson & Bishop [4], we saw that the tests used here were reliable in the non-autistic sample, and we assessed whether this was also the case among the autistic adults using the same approaches. We report a summary of item-level statistics for the tests, as well as reliability coefficients: Cronbach's alpha (with 95% CIs), standard error of measurement and Revelle's beta. We used item response (IRT) models to test the extent to which all items on a test tap a single latent ability (i.e. are unidimensional) and report the root mean square error of estimation (RMSEA) for each model. Using IRT modelling, we produced item characteristic curves to check item quality and planned to exclude any items with low, flat curves reflecting chance-level accuracy across the ability spectrum. IRT models were also used to generate the measured variables for the Awkward Dialogues and the Frith-Happé Animations (as described in Measures). For this analysis, R packages psych [22] and mirt [23] were used.

Outlying scores on the tests were identified using the method of Hoaglin & Iglewicz [24], such that scores 2.2 times the interquartile range below the lower quartile were excluded from the dataset. On the Implicature Comprehension Test, we excluded any test scores where the individual was an outlier on the explicit-response items; on the Awkward Dialogues, we excluded scores where at least one trial was invalid (see Measures for details); these exclusions were made to remove scores where the individual had not engaged well with the tasks. As the factor analyses could account for missing scores, exclusion of individual scores did not mean exclusion of that participant from the dataset, though we checked for the sensitivity of the models where individuals had several scores excluded.

We inspected the data for normality and multivariate outliers using R package MVN [25]. Multivariate outliers were defined as individuals whose adjusted Mahalanobis distance was above the 97.5th percentile of the chi-distribution. Maximum-likelihood estimation was used for the factor analysis. As this is based on the multivariate normal distribution, it was important to check whether the data departed from this distribution [26], although given that robust estimation was used in the factor analysis, assumptions of normality were relaxed [27]. Nonetheless, in the case of multivariate outliers being present, we planned to transform variables using the Tukey ladder of power transformations using R package rcompanion [28] to reduce skew, and then test for multivariate outliers again. Remaining outliers would be excluded. The sensitivity of the analysis to any data transformation or outlier exclusion was evaluated by comparing results based on transformed and non-transformed data with and without outliers included.

### 2.5.2. Confirmatory factor analysis: hypothesis 1 (identifying factor structure)

Using confirmatory factor analysis, we aimed to replicate a two-factor correlated traits model of communication skills already established in non-autistic adults, in a larger sample including autistic adults alongside the other adults. For this model, four variables were set to load on the 'social understanding' factor and two variables on the 'core language' factor, as set out in Measures. In line with the pre-registered analysis described on OSF, we planned to test whether this model fitted the data better than a one-factor model with all tests loading on one general factor. However, as reported in Results, there seemed to be some mis-specification in the pre-registered models, as the vocabulary measure seemed quite distinct from other variables. Therefore, a new one-factor model of 'receptive communication' excluding Receptive Vocabulary was also tested. All factor analyses were run using with R package lavaan [29] with full information likelihood estimation to deal with missingness and robust standard errors to allow for non-normality. Confirmatory fit indices (CFIs) and RMSEA with 90% confidence intervals are presented for all models.

### 2.5.3. Confirmatory factor analysis: hypothesis 2 (testing for group differences)

The plan pre-registered on OSF was to test for group differences on the two-factor model. However, as indicated above, there was some mis-specification in the two-factor model and a revised one-factor model of 'receptive communication' accounted better for the relationships between tests. Therefore, this revised one-factor model was analysed here. We tested the hypothesis that autistic individuals had lower scores on the communication tests, by specifying a MIMIC (multiple indicators multiple causes) model, with autism diagnosis as a covariate at the factor level. We compared a model in which the regression path from the diagnosis covariate to the 'receptive communication' factor was set to zero and a model in which the path was allowed to vary using a Satorra–Bentler scaled difference chi-square test. Essentially, this tested whether there was a significant difference between autistic and non-autistic people on the 'receptive communication' factor. The magnitude of the standardized regression path gave an effect size for the difference. As part of this analysis, we also tested the measurement invariance of the two-factor model across autistic and non-autistic groups, i.e. whether the same factor structure was present across the groups. This involves modelling the two groups separately before testing for metric, scalar and strict measurement invariance by progressively fixing the factor loadings, indicator intercepts and residuals such that they are constrained to be the same across groups [30]. As each of these more stringent models was run, a chi-square difference test was used to assess whether fit significantly deteriorated.

### 2.5.4. Confirmatory factor analysis: hypothesis 3 (investigating variability in the autistic group)

We hypothesized that performance on the language/communication test battery would relate to self-reported and researcher-rated global communication skills in face-to-face interaction. For this analysis, we re-ran the 'receptive communication' one-factor model but only used data collected from the autistic adults, with regression paths specified between the factor and self-reported communication challenges (Communication Checklist – Self Report (CC-SR) pragmatic composite) and researcher-rated communication (ADOS-2 total). We report the magnitude of the standardized regression paths.

## 3. Results

We had previously shown that the test battery measures were reliable in a general population sample of adults. The same seems to be true when used with adults who have clinical diagnoses of autism, as shown by the reliability coefficients and item-level statistics in table 1. Table 1 also shows the RMSEA for IRT models, which indicate how well unidimensional models describe the tests. The Test of Fillers and Backchannels had a high RMSEA, suggesting that this test was multidimensional; however, this is in the context of good reliability coefficients, so it does not seem a cause for concern. We examined item characteristic curves for all items on all tests, and these showed steep curves, reflecting good item quality. As such, there was no reason to adapt the content of the tests.

Table 2 shows descriptive statistics for each of the measures by group (autistic and non-autistic). The measure of general cognitive ability (the ICAR Sample Test) was only administered to the autistic adults. Mean score was 8.16 (s.d. = 3.80). This is essentially identical to the ICAR sample described above ($M$ = 8.21, s.d. = 3.77), suggesting that the cognitive ability of the autistic group was similar to the general

**Table 1.** Reliability analysis, including Cronbach's alpha and 95% confidence intervals, standard error of measurement (SEm), Revelle's beta and IRT RMSEA for a unidimensional model for each test. We also present item-level statistics summarized by quartiles for corrected item-total correlations (totals excluding the item) and item-level accuracy.

| test | alpha | alpha 95% CIs | SEm | beta | IRT RMSEA | item-total correlations | | | item-level accuracy | | |
|---|---|---|---|---|---|---|---|---|---|---|---|
| | | | | | | Q1 | Q2 | Q3 | Q1 | Q2 | Q3 |
| implicature comprehension test | 0.89 | 0.87, 0.91 | 2.14 | 0.83 | 0.02 | 0.35 | 0.42 | 0.48 | 0.62 | 0.75 | 0.85 |
| test of fillers and backchannels | 0.85 | 0.82, 0.88 | 3.05 | 0.87 | 0.11 | 0.27 | 0.32 | 0.42 | 0.49 | 0.62 | 0.82 |
| receptive grammar | 0.94 | 0.93, 0.95 | 1.39 | 0.86 | 0.02 | 0.40 | 0.49 | 0.56 | 0.70 | 0.84 | 0.89 |
| receptive vocabulary | 0.81 | 0.77, 0.85 | 2.32 | 0.68 | 0.03 | 0.28 | 0.35 | 0.43 | 0.35 | 0.56 | 0.65 |
| test of local textual inference | 0.78 | 0.73, 0.82 | 1.94 | 0.64 | 0.02 | 0.24 | 0.35 | 0.40 | 1.58 | 1.75 | 1.83 |

**Table 2.** Descriptive statistics.

| | N | mean | s.d. | min | max | skew | kurtosis |
|---|---|---|---|---|---|---|---|
| **autistic adults** | | | | | | | |
| implicature comprehension test | 66 | 22.30 | 6.46 | 10 | 35 | 0.10 | −0.82 |
| control items for implicature comprehension test | 66 | 9.20 | 0.93 | 7 | 10 | −0.84 | −0.41 |
| test of fillers and backchannels | 71 | 23.18 | 7.87 | 2 | 36 | −0.46 | −0.45 |
| awkward dialogues | 68 | −0.08 | 0.76 | −1.68 | 1.02 | −0.18 | −0.85 |
| Frith-Happé animations | 71 | −0.10 | 0.85 | −2.90 | 1.53 | −0.38 | 0.15 |
| control items for Frith-Happé animations | 71 | 9.17 | 2.43 | 3 | 12 | −0.99 | 0.17 |
| receptive vocabulary | 71 | 14.49 | 5.33 | 0 | 25 | −0.48 | −0.3 |
| receptive grammar | 63 | 34.08 | 5.69 | 19 | 41 | −0.96 | 0.10 |
| test of local textual inference | 68 | 33.24 | 4.13 | 22 | 40 | −0.82 | 0.06 |
| **non-autistic adults** | | | | | | | |
| implicature comprehension test | 118 | 28.92 | 4.49 | 8 | 36 | −1.20 | 2.80 |
| control items for implicature comprehension test | 118 | 9.64 | 0.64 | 7 | 10 | −1.70 | 2.39 |
| test of fillers and backchannels | 120 | 26.84 | 5.93 | 7 | 37 | −0.83 | 0.79 |
| awkward dialogues | 119 | 0.07 | 0.74 | −1.87 | 1.02 | −0.59 | −0.46 |
| Frith-Happé animations | 120 | 0.08 | 0.71 | −1.86 | 1.53 | −0.27 | −0.27 |
| control items for Frith-Happé animations | 120 | 9.69 | 2.16 | 2 | 12 | −1.15 | 1.04 |
| receptive vocabulary | 120 | 12.10 | 4.45 | 3 | 24 | 0.31 | −0.26 |
| receptive grammar | 119 | 35.04 | 4.45 | 19 | 44 | −0.75 | 0.47 |
| test of local textual inference | 115 | 34.43 | 3.24 | 24 | 40 | −0.73 | 0.30 |

population. Table 3 shows estimates of group difference (Cohen's $d$) on each test. We calculated group differences based on two different definitions of the autistic group: anyone reporting a clinical diagnosis in the first case and only those meeting the 'autism spectrum' cut-off on the ADOS-2 in the second. The Cohen's $d$ results presented in table 3 show a substantial difference on the Implicature items, a medium difference on the Test of Fillers and Backchannels, a small difference on the Frith-Happé Animations, all in favour of the non-autistic group, and a medium difference on Receptive Vocabulary in favour of the autistic group. Other tests show smaller differences, with confidence intervals spanning zero, though the trend is for higher scores in the non-autistic group.

## 3.1. Confirmatory factor analysis: hypothesis 1 (identifying factor structure)

The one-factor model incorporating the various 'social understanding' tests alongside Receptive Grammar and Receptive Vocabulary did not fit the data well, CFI = 0.76, RMSEA = 0.15, 90% CI [0.11, 0.20]. However, when running the two-factor model with separate 'social understanding' and 'core language' factors, there were also issues with model fit, as the residual variance for Receptive Grammar was negative. This therefore represented a Heywood case, suggesting the model was mis-specified [31]. On inspection of the correlation matrix (table 4), we see that Receptive Grammar is inter-related with all the variables, whereas Receptive Vocabulary is moderately related to Receptive Grammar but little else, suggesting the language and communication measures clustered together whereas vocabulary was rather distinct.

**Table 3.** Cohen's *d*, showing magnitude of the difference between the autistic and non-autistic groups on each measure. Negative values indicate lower performance. We present effect sizes comparing all participants reporting a clinical diagnosis ($N =$ 71) and the non-autistic group ($N = 120$) in the left-hand columns. On the right, the autistic group comprises those reporting a diagnosis who met ADOS-2 criteria for 'autism spectrum' or 'autism' ($N = 55$); the non-autistic group was the same ($N = 120$).

| | autistic group (diagnosis reported) | | | autistic group (diagnosis AND ADOS-2 criteria met) | | |
| --- | --- | --- | --- | --- | --- | --- |
| | estimate | lower 95% CI | higher 95% CI | estimate | lower 95% CI | upper 95% CI |
| implicature comprehension test | −1.25 | −1.58 | −0.92 | −1.16 | −1.51 | −0.80 |
| test of fillers and backchannels | −0.54 | −0.85 | −0.24 | −0.56 | −0.89 | −0.24 |
| awkward dialogues | −0.20 | −0.51 | 0.10 | −0.27 | −0.60 | 0.05 |
| Frith-Happé animations | −0.24 | −0.54 | 0.05 | −0.36 | −0.69 | −0.04 |
| control items for Frith-Happé animations | −0.23 | −0.53 | 0.07 | −0.25 | −0.57 | 0.07 |
| receptive vocabulary | 0.50 | 0.20 | 0.80 | 0.51 | 0.18 | 0.83 |
| receptive grammar | −0.20 | −0.50 | 0.11 | −0.24 | −0.58 | 0.10 |
| test of local textual inference | −0.33 | −0.63 | −0.03 | −0.20 | −0.54 | 0.13 |

**Table 4.** Correlations between language tests included in the factor analysis using pairwise-complete observations. Variables have been transformed. Textual inference = Test of Local Textual Inference.

| | fillers and backchannels | awkward dialogues | Frith-Happé animations | receptive vocabulary | receptive grammar | textual inference |
| --- | --- | --- | --- | --- | --- | --- |
| implicature comprehension test | 0.41 | 0.30 | 0.31 | −0.05 | 0.31 | 0.30 |
| test of fillers and backchannels | | 0.21 | 0.31 | 0.12 | 0.32 | 0.25 |
| awkward dialogues | | | 0.24 | 0.16 | 0.31 | 0.29 |
| Frith-Happé animations | | | | 0.22 | 0.26 | 0.25 |
| receptive vocabulary | | | | | 0.46 | 0.16 |
| receptive grammar | | | | | | 0.26 |

Based on the correlation matrix, we specified a new one-factor model of 'receptive communication' excluding Receptive Vocabulary (which can be seen in figure 1). This new one-factor model gave an excellent fit to the data, CFI = 1, RMSEA = 0.00, 90% CI [0.00, 0.05]. This analysis was run using variables transformed to better approximate normal distributions, as there were several cases that were multivariate outliers in the untransformed, but not the transformed, data. However, results were essentially the same when run with transformed or non-transformed data. Note that the 'receptive communication' factor did not simply represent general cognitive ability; the correlation between these factor scores and performance on the ICAR Sample Test in the autistic group was only 0.30.

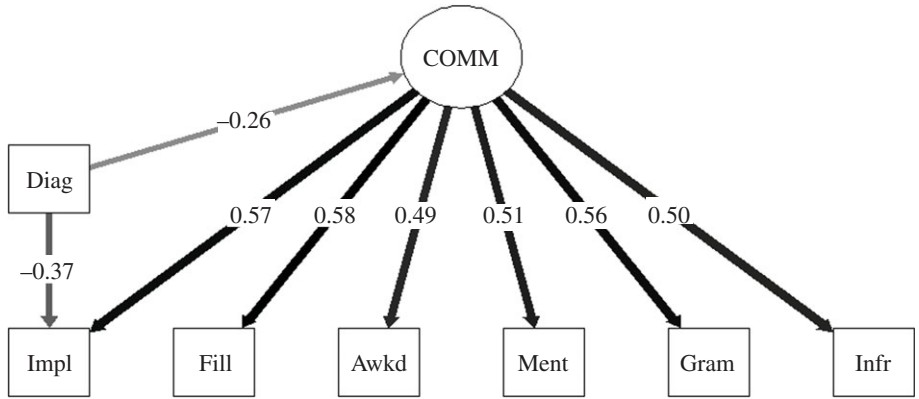

**Figure 1.** One-factor model of 'receptive communication' skills across autistic and non-autistic adults, incorporating diagnosis (of autism) included as a covariate. COMM = 'receptive communication' factor; Diag = Diagnosis of autism; Impl = Implicature Comprehension Test; Fill = Test of Fillers and Backchannels; Awkd = Awkward Dialogues; Ment = Mental state attribution on the Frith-Happé Animations; Gram = Receptive Grammar; Infr = Test of Local Textual Inference.

**Table 5.** Measurement invariance testing using multi-group confirmatory factor analysis across autistic and non-autistic groups.

|  | model Df | model chi-square | chi-square difference | Df difference | p-value | comparison | CFI | RMSEA |
|---|---|---|---|---|---|---|---|---|
| configural | 18 | 10.97 |  |  |  |  | 1 | 0.00 |
| metric | 23 | 12.11 | 1.12 | 5 | 0.952 | configural | 1 | 0.00 |
| scalar | 27 | 16.20 | 4.10 | 4 | 0.393 | metric | 1 | 0.00 |
| strict | 33 | 18.28 | 2.52 | 6 | 0.866 | scalar | 1 | 0.00 |

## 3.2. Confirmatory factor analysis: hypothesis 2 (testing for group differences)

Diagnosis was added as a covariate to the one-factor 'receptive communication' model. We compared models where the regression path from diagnosis to the factor was allowed to vary and where it was set to zero. The model with the free regression path fitted the data significantly better, $\chi_1^2 = 12.44$, $p < 0.001$, indicating that autistic and non-autistic groups did differ on the factor. However, the fit statistics of this model were relatively weak, CFI = 0.88, RMSEA = 0.10, 90% CI [0.06, 0.13]. Examining the pattern of residuals for the model, it seemed that poor fit was because group differences were only specified at the factor level. As can be seen in table 3 above, the pattern of differences across groups was not uniform across tests. While differences were generally subtle, there was a very considerable difference on the Implicature Comprehension Test. To better reflect this, we also added a regression path to the model between diagnosis and this test. Following the refinement, the model showed much improved fit statistics, CFI = 1, RMSEA = 0.00, 90% CI [0.00, 0.05]. The magnitude of the standardized path between diagnosis and the 'receptive communication' factor was small in size, $\beta = -0.26$, $p = 0.005$, reflecting somewhat lower scores in the autistic group. In addition to this domain-level difference, there was a small–medium difference on the Implicature Comprehension Test, with non-autistic participants scoring higher, $\beta = -0.37$, $p < 0.001$.

One assumption of testing for differences across a factor model is that factors are derived in the same way across groups. Therefore, we examined the extent to which this was the case across the autistic and non-autistic groups by progressively fixing loadings, intercepts and error terms of the indicators across groups. Clearly, we cannot expect the intercept for the Implicature Comprehension Test to be the same, as there were differences in this test not accounted for by differences in factor scores. In the analysis for measurement invariance, we therefore allowed the intercept for this test to vary, but progressively fixed all other parameters. As can be seen in table 5, fit did not significantly deteriorate as parameters were fixed, indicating that the factors were measured in a similar way across groups.

To aid in interpretation we considered how results were influenced by (i) restricting the autistic group to individuals meeting ADOS-2 criteria, and (ii) taking gender into account. With respect to the composition of the autistic group, we ran the same factor analysis described above but this time

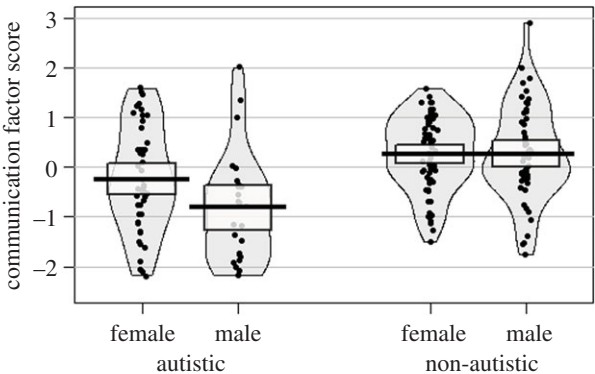

**Figure 2.** Pirate plot showing 'receptive communication' factor scores by group (autistic and non-autistic) and gender. Central tendency is mean with 95% CI. Scores across the sample have been standardized to have a standard deviation of 1.

we only included autistic adults meeting ADOS-2 criteria ($N = 55$) and the non-autistic comparison group ($N = 120$). Results were very similar for this model, CFI = 1, RMSEA = 0.00, 90% CI [0.00, 0.08]. Regression pathways were also very similar between diagnosis and the 'receptive communication' factor, $\beta = -0.30$, $p = 0.001$, and performance on the Implicature Comprehension Test, $\beta = -0.27$, $p = 0.005$. Moving on to gender, we tested whether autistic men and women scored differently on the 'receptive communication' factor by running a two-way ANOVA on factor scores with a group (autistic or non-autistic) and gender as between-subjects variables. There was no main effect of gender, $F_{1,186} = 0.88$, $p = 0.350$, but there was a marginally significant group by gender interaction, $F_{1,186} = 3.92$, $p = 0.049$. While this analysis is exploratory and the sample size was not powered for subgroup analysis by gender, an inspection of figure 2 would suggest a trend for autistic men to score lower on the 'receptive communication' factor compared with autistic women, with no gender difference in the non-autistic group. As well as this difference on the test battery, it should also be noted that autistic men and women scored quite differently on the researcher-rated measure of global communication, the ADOS-2, $t_{39.87} = 5.19$, $p < 0.001$, Cohen's $d = 1.41$, with women showing fewer features characteristic of autism. There were, however, no gender differences in general ability, $t_{44.75} = 0.18$, $p = 0.856$, or self-reported communication difficulties on the CC-SR pragmatic scale, $t_{44.15} = 0.26$, $p = 0.795$. These gender-related communication trends will need to be investigated in future research.

## 3.3. Confirmatory factor analysis: hypothesis 3 (investigating variability in the autistic group)

For this final factor analysis, we moved from the comparison of the autistic and non-autistic groups to assess whether scores of autistic people on the test battery were predictive of variability in global communication skills, as rated by the researcher on ADOS-2, or as self-rated on the CC-SR pragmatic composite. Mean score on the ADOS-2 was 9.65 (s.d. = 3.91) and on the CC-SR pragmatic composite was 23.76 (s.d. = 14.60); in terms of norm-referenced z-scores, this translates to an average CC-SR score of −2.27 (s.d. = 1.61). We included regression paths between the 'receptive communication' factor and researcher-rated face-to-face communication on the ADOS-2 and self-reported communication challenges on the CC-SR pragmatic scale. Fit was excellent in this model run in the autistic group, CFI = 1, RMSEA = 0.00, 90% CI [0.00, 0.08]. There were moderate-sized regression paths between the 'receptive communication' factor and ADOS-2 score, $\beta = -0.43$, $p < 0.001$, and CC-SR pragmatic score, $\beta = -0.40$, $p = 0.001$. This analysis suggests that the test battery did pick up difficulties related to face-to-face interaction. It is possible that relationships between these measures might have been simply due to general cognitive ability, so to rule out this possibility, we regressed ICAR Sample Test scores on the ADOS-2 and CC-SR pragmatic totals, extracted the residuals for these two variables, and re-ran the factor analysis using residualized scores (thereby controlling for general ability). Regression paths remained significant for the ADOS-2, $\beta = -0.37$, $p = 0.005$, and CC-SR pragmatic scale, $\beta = -0.28$, $p = 0.047$. Finally, it is interesting to note that there was no correlation between the ADOS-2 and CC-SR pragmatic scale, indicating that researcher-rated and self-reported communication skills represented entirely different constructs, $p = 0.309$. See a path diagram in figure 3.

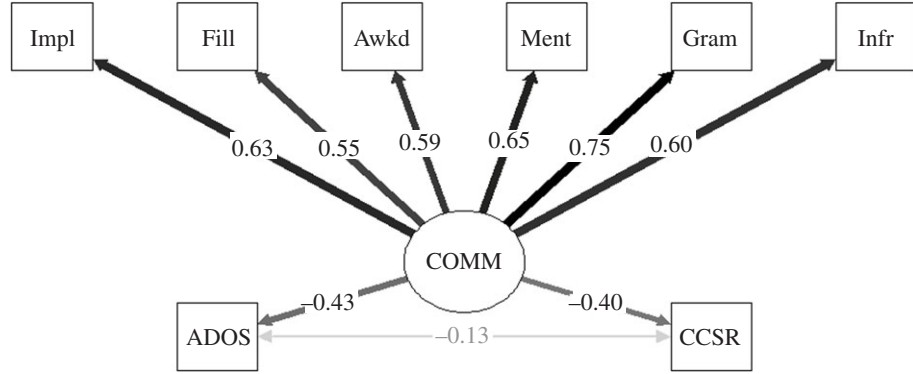

**Figure 3.** One-factor model of 'receptive communication' skills in autistic adults, showing factor scores extracted from the test battery as a predictor of global communication ratings. COMM = 'receptive communication' factor; Impl = Implicature Comprehension Test; Fill = Test of Fillers and Backchannels; Awkd = Awkward Dialogues; Ment = Mental state attribution on the Frith-Happé Animations; Gram = Receptive Grammar; Infr = Test of Local Textual Inference; ADOS = ADOS-2 total; CCSR = CC-SR pragmatic composite.

## 4. Discussion

The results raise challenging questions about the nature of communication difficulties in autism. While we found that autistic people underperformed on the language and communication test battery, it was clear that the subtle group differences on the tests were *not* well accounted for by a specific 'social understanding' factor relating to 'theory of mind' as hypothesized. Instead, the root seemed to be more domain-general processing differences. Also, we expected any difficulties on tests to reflect a single underlying difference between groups, but the reality was less simple. In addition to a domain-level processing difference, there seemed to be further differences in cognitive style/processing present on one particular test, the Implicature Comprehension Test, as there was a much greater difference in scores between autistic and non-autistic people on this than on any other test, as shown in figure 1. This supports the idea of multiple influences affecting the communication phenotype in autism. Finally, we expected that performance on the test battery would predict global communication difficulties in face-to-face scenarios, and while this was the case, there was an unexpected dissociation between observed and self-reported difficulties, and this adds further complexity to the communication phenotype in autism, as discussed below.

### 4.1. Comparing autistic and non-autistic adults: a domain-level difference in information-processing

First, it should be noted that the hypothesized structure of communication skills measured by the test battery was not supported. We expected the pattern of scores across the tests to cluster into two factors: 'social understanding', representing more complex inferential skills required in processing social meaning and intentions, and 'core language' representing formal aspects of language like grammar and vocabulary. Although vocabulary was distinct from other tests and only showed a clear relationship with grammar, all other language and communication measures (including grammar) were inter-related. Therefore, the most appropriate factor analysis accounting for patterns across the test battery was a single-factor model representing a broad domain in 'receptive communication' skills (but excluding vocabulary). Autistic people underperformed on this domain.

However, what precisely did that domain represent? The test battery was devised to put special demand on the ability to make inferences about social information. This included inferences about implied meanings in conversation, social functions of conversational fillers, awkwardness in conversation and social scenarios represented in abstract scenarios. However, these tests clustered with tests requiring the individual (i) to judge the grammaticality of isolated sentences, and (ii) to process narrative meaning that carried from one sentence to the next (without the need for interpreting behaviour or emotions). The common denominator in all these tests was less a requirement to make judgements about conversations and social situations, but more generally to make *normative judgements about novel communicative stimuli*. We say 'communicative stimuli' rather than 'language',

since the Frith-Happé Animations involves non-verbal social situations. 'Novel' is used as a qualifier because all the tests involved stimuli the participant had not encountered before; they could not rely on a learned response to a familiar stimulus. It should be noted that vocabulary did not cluster with these tests, potentially because vocabulary skills do depend on learning and familiarity, i.e. crystalized word-level knowledge. Notably, autistic people sampled in this study had a particular strength in this area. Finally, we should note that judgements were 'normative'; there were no right or wrong answers based on the application of clear-cut rules. For tests involving judgements about social situations or implied meaning, it is clear we are guided by our expectation of how other people might interpret the stimuli, but this is probably the case for all the tests. For example, in the grammar test, incorrect items were devised to sound clumsy rather than to violate explicit rules, and so consideration of the typical native speaker's reaction to items was probably a feature of the test.

As noted above, autistic people showed a reduced tendency to make normative judgements about the communicative stimuli included in the cluster of tests represented by the 'receptive communication' factor, and their scores on this factor were a moderate predictor of both self-reported and researcher-rated communication difficulties. How might we understand these findings in the context of leading cognitive accounts of autism? We could look to information-integration accounts, such as the 'weak central coherence' hypothesis that proposes autistic people favour local, detailed processing over global processing [5], and the 'complex information-processing' model that proposes autistic people can struggle with assimilating information, especially when new or more subtle forms of organization are needed [32,33]. Under these accounts, autistic people are less likely to integrate features in context, and this would predict the difficulties observed here on tests requiring inferences, e.g. about implied meaning or narrative, etc. (for a review of inferencing difficulties in autism, see [34]). These accounts are also consistent with the dissociation found in this study between word-level abilities (which were a strength for autistic people) and tasks that involved processing meaning in context at the sentence level and above (on which autistic people tended to underperform). Overall, our findings agree with studies of language comprehension influenced by 'weak central coherence' theory, which have reported reduced processing of overall coherence both 'locally' at the sentence level and 'globally' at the discourse level among autistic people [35,36].

## 4.2. No evidence for an isolated 'theory of mind' difficulty

In this study, we hypothesized that any difficulties on the tests could be attributed to 'theory of mind' (i.e. reasoning about mental states), and on the face of it, we could link a difficulty in making normative judgements about communicative stimuli to the highly influential 'theory of mind' account of autism [7]. However, there are a couple of reasons to be wary of this explanation. First, it does not seem plausible that the 'receptive communication' factor represented 'theory of mind'. While the Frith-Happé Animations and Awkward Dialogues could be viewed as 'theory of mind' tests, it is difficult to see how we could say the same about tests of grammar and narrative-based inferencing without overextending the definition of 'theory of mind' in unsatisfactory ways; all these tests showed similar loadings on the factor, so it seems unlikely it represented 'theory of mind'. In addition, it is worth noting that group differences on most of the tests were relatively subtle. This indicates that autistic people clearly can perform reasonably well on tests with a social reasoning element, such as the Frith-Happé Animations, while simultaneously experiencing substantial difficulties in embodied social interaction and day-to-day life. While performance on the test battery did predict self- and researcher-rated social communication difficulties, we should also not ignore the significant discrepancy between scores on the tests (slightly below average, at the group level) and the very considerable challenges the autistic people experienced in active social scenarios (as indicated by clinical-range scores on the ADOS-2 and CC-SR). This discrepancy has been noted before (e.g. [37]) and suggests that we should not assume that explicit social reasoning and action in the social world rely on precisely the same cognitive substrates or have the same developmental trajectories.

Before moving on, it is worth lingering on the Frith-Happé Animations. This was the test that most explicitly targeted 'theory of mind', and on which we might expect autistic people to show a particular dip in performance under the 'theory of mind' account of autism. However, this was not the case. Autistic people performed very similarly on both the 'theory of mind' and control items, and any group differences on either set of items only represented a small effect size. This result does depart somewhat from previous research, which has on average found a medium effect size difference between autistic and non-autistic people on the 'theory of mind' items (see [38] for a meta-analysis). This difference from previous research might have been due to the online methods used in this study

and the nature of the sample recruited: these two factors suggest we should not view 'theory of mind' as a universal, invariant difficulty for autistic people, but that performance on 'theory of mind' tasks is dependent on context and individual differences. Whereas the Frith-Happé Animations is typically administered face-to-face, a remote online administration was used here, which may have facilitated performance among autistic people. Certainly, among children, a study reported that differences between autistic and non-autistic groups emerged on a researcher-administered 'theory of mind' task but not a computerized version [39]. With respect to the nature of the sample, there were more women than men (unusually for autism research involving this task) and women performed significantly better than men on the Frith-Happé Animations, Cohen's $d = 0.80$, 95% CI [0.29, 1.31]; this may have led to smaller overall differences between autistic and non-autistic people than are typically found on the task (as reported in [38]). On the other hand, it should be noted that our study *is* consistent with existing research insofar as other studies also do *not* find a clear-cut difference between performance on the 'theory of mind' and control items of the Frith-Happé Animations among autistic people [38], even though the original study reported a considerable difference specifically on the 'theory of mind' items [40]. This is in line with other research that has failed to replicate a specific 'theory of mind' difficulty among autistic people and/or has questioned the construct validity of 'theory of mind' (for a review, see [9]). All the same, we should be careful not to deny the usefulness of the 'theory of mind' construct in understanding autism, as links have been well-documented [7,8]. The critical point might be that weaknesses with 'theory of mind' are not inevitable, but are likely to interact with social experience over development [41] and be influenced by other aspects of cognitive development (as noted in the Introduction, a broad set of skills touch on 'theory of mind'). This developmental perspective could be considered in future longitudinal research assessing young people with age-appropriate versions of tasks used in the present study.

## 4.3. Multiple influences on the communication phenotype in autism

There was a spiky profile of group differences on the test battery. Autistic people scored higher on vocabulary but substantially lower than non-autistic people on the Implicature Comprehension Test in the context of relatively small differences in favour of non-autistic people on other tests. The large difference in performance on the Implicature Comprehension Test was partly explained at the factor level—autistic individuals scored lower on the 'receptive communication' factor, which incorporated the implicature test. However, when attempting to explain all group differences at the level of this factor, the model fit was poor. Instead, we needed to allow for a further difference at the level of the implicature task too. What this shows is that multiple influences affected the performance of autistic people on the task. This suggestion agrees with the idea that features of autism are 'fractionable'— that there are multiple distinct influences on the autism phenotype [42]. With respect to the Implicature Comprehension Test, some of the group difference on this task was 'fractionated' at the general factor level, and in explaining the nature of that difference, we appeal to a domain-general information-integration account. In Wilson & Bishop [43], we provided evidence that a cognitive preference for certainty and explicit communication played a role in performance on the task, so perhaps the group difference 'fractionated' at the specific test-level reflected that cognitive preference. This is speculative, but suggests we ought to move away from a cognitive model that suggests there is one source of the communication phenotype in autism.

In addition to a comparison of communication in autistic and non-autistic people, we investigated variability in communication among autistic people. For this, we used three methods: the test battery already discussed, face-to-face interaction with a researcher and self-reported difficulties. We expected these measures to converge, with individuals falling along a single continuum of communication ability. This was not supported by the data; instead, communication is multifaceted, and it is clearly necessary to use multiple measures to characterize the communication profile of autistic people. Most individuals showed clinically significant communication difficulty during one-to-one social interaction, i.e. over 80% scored 7 or above on the ADOS-2. This converts to a score below the 2nd percentile, based on general population epidemiological data reported in Brugha *et al.* [44]. However, the extent to which this researcher-rated difficulty co-occurred with (i) difficulties on the language/ communication test battery, and (ii) self-reported challenges in day-to-day communication was highly variable.

Factor scores extracted from the test battery showed moderate overlap with the self-reported and researcher-rated measures of global communication skills (with standardized coefficients of 0.40 and 0.43), indicating that it is possible to capture some of the difficulty experienced by autistic people in

day-to-day conversation using computerized comprehension tests. However, there is clearly more to communication than what could be measured by these tests. That is, perhaps, unsurprising. What is more striking is that the two measures of global communication skills (ADOS-2 and CC-SR) were not correlated. This means that the extent of autistic communication behaviours observed by a researcher (or clinician) in a one-off interaction will say little about how a person experiences communication difficulty in day-to-day life. It is likely that the types of challenges that individuals are self-reporting on the CC-SR come and go; a communication mishap might happen once a day, perhaps quite unpredictably and more often in stressful situations, and this could be easily missed in an ADOS-2 assessment, which just samples a moment in time in a formal context. However, it is unlikely that the ADOS-2 simply 'misses' difficulties; for one thing, most individuals scored in the clinically significant range on this measure, as they did on the CC-SR. Instead, we need to view self-reported and researcher-rated communication difficulties as different constructs. The ADOS-2 requires the administrator to create a social environment designed to bring out some of the challenges faced by autistic people and to make judgements about aspects of a participant's social communication that are characteristic of autism, guided by an awareness of how other researchers/clinicians would use the measure (as ADOS-2 training requires the administrator to demonstrate a high level of consensus with other users of the ADOS-2). The CC-SR, as a self-report questionnaire, measures an individual's perceptions of communication challenges they may have. While the CC-SR attempts to quantify such challenges objectively by asking participants how frequently a certain aspect of communication poses a problem for them, the questionnaire nonetheless relies on an individual noticing a problem, conceptualizing it and reporting it as such, and an individual's perceptions of their communication experiences might or might not square with other people's observations. Besides frequency of actual communication challenges, scores on the CC-SR are going to be affected by an individual's insight, cognitive biases (such as overly negative appraisals) and reporting biases (such as social desirability, or tendency to report problems and seek help). The phenomenon of 'masking', where an individual is aware of having social difficulties and adopts strategies in social situations to try mitigating those difficulties (e.g. [45]), is also likely to impact on how difficulties are reported and manifest in social situations. This plethora of factors indicates that the relationship between observed difficulties and perceptions of difficulty is likely to be complex, and so the lack of correlation between ADOS-2 and CC-SR scores need not reflect issues with the validity of the measures, but instead shows the complexity of communication experiences for autistic individuals. The very fact that the communication measures did not indicate a single continuum of communication skills suggests that difficulties do not occur in all contexts, which is important to bear in mind when making autism-positive environments.

We should also consider autistic strengths and not necessarily assume weaknesses in all aspects of communication. In this respect, it is worth drawing attention to a strength we observed in the autistic group relative to the non-autistic group: vocabulary. This was not predicted and is perhaps a result of recruiting people to a study focused on language and communication: perhaps individuals with a special interest in words and language were motivated to volunteer, and such special interests may have been more common in the autistic group. Of course, this is merely speculation, but it is important not to erase autistic strengths to focus single-mindedly on what might be considered 'weaknesses'.

## 4.4. Gender and communication skills in autistic adults

While we did not explicitly set out to test for differences between autistic men and women in communication skills, the exploratory analysis suggested that gender might play an important role. Autistic women showed less communication difficulty, both on the cognitive test battery and during one-off interaction, but reported similar day-to-day communication challenges as men and were of comparable general cognitive ability. In the past, greater levels of impairment and lower cognitive ability tended to be reported for females on the spectrum [46], although this is probably because of diagnostic biases, with clinicians mainly identifying autistic presentations in males and only diagnosing autism in females with the most obvious difficulties. There is an increasing appreciation that females may be particularly likely to mask some of their difficulties using learned social strategies (e.g. [45,47]). As such, girls may show more reciprocal conversation during diagnostic assessments [48] and women have been observed as showing more 'normative' social behaviours on the ADOS-2 [14], which was replicated in the present study. There has been little investigation of gender differences in language abilities of autistic individuals. One small study ($N = 52$, with equal numbers of autistic and non-autistic boys and girls) found that autistic girls tended to have a subtly different language profile to boys [49].

On tests of figurative language, inferencing and word associations, autistic children tended to underperform, with autistic girls scoring higher than boys, though this was in the context of a gender effect in the non-autistic children too. In the present project, autistic women tended to perform better on the 'receptive communication' factor than autistic men, and this seemed to be specific to autistic people rather than reflecting a gender difference also present in the general population. For autistic women, language skills represent an area of relative strength that might, for instance, be useful in compensating for social difficulties. However, the flipside is that clinicians may be expecting to see communication difficulties, and where these don't manifest during testing, there may be a tendency to underestimate the challenges that some autistic women experience in their day-to-day life. Having noted these average trends, it is also important to underscore the substantial variability found in autistic people of all genders.

# 5. Conclusion

In confirmatory factor analysis, we found that individuals with an autism spectrum diagnosis scored lower on a 'receptive communication' factor comprising six tests requiring the individual to make normative judgements about novel communicative stimuli. While a domain-level difference between autistic and non-autistic people was expected, this result was contrary to the hypothesis as we expected these group differences would represent 'social understanding' or 'theory of mind', but it seemed this factor was rather more domain-general and involved making inferences and judgements about a full range of uses of language, including those that were not explicitly embedded in a live social context. Skills in this area seemed to have a real-world implication for autistic adults, as scores on the 'receptive communication' factor predicted difficulties in face-to-face interaction. A final point to underscore is variability: the group difference between autistic and non-autistic adults was subtle, and there was a full range of variability in both groups. Adding to this picture of complexity was the lack of a relationship between self-reported and observed difficulties in social interaction among autistic people. This underscores the importance of any communication support for autistic people being tailored to their specific strengths and difficulties.

Ethics. Ethical review for this study occurred in two stages, separately for the autistic and non-autistic participants, as the groups were recruited sequentially. The first stage of the study was granted ethical clearance in July 2018 (Ref. R57087/RE002) and the second in November 2018 (Ref. R59912/RE002) by the Medical Sciences Division of the Oxford University Research Ethics Committee. All participants gave informed consent to participate in the study.
Data accessibility. Data and scripts are accessible on the Open Science Framework (https://osf.io/b9t8a/).
Authors' contributions. A.C.W. was involved in conceptualization, production of materials, investigation, data curation, formal analysis and writing the manuscript. D.V.M.B. was involved in conceptualization, formal analysis, manuscript editing, supervision and funding acquisition. All authors gave final approval for publication.
Competing interests. We declare we have no competing interests.
Funding. This research was funded by the European Research Council (Ref. 694189).
Acknowledgements. Our warmest thanks to all the participants who took part in this research, and thank you to Autistica for helping us reach out to the autism community during our recruitment phase.

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
