## [Reviewer comments · Royal Society Open Science]

Review History

RSOS-200845.R0 (Original submission)

Review form: Reviewer 1 (Sue Fletcher-Watson)

Is the manuscript scientifically sound in its present form?

Yes

Are the interpretations and conclusions justified by the results?

Yes

Is the language acceptable?

Yes

Do you have any ethical concerns with this paper?

No

Have you any concerns about statistical analyses in this paper?

No

Recommendation?

Accept with minor revision (please list in comments)

Comments to the Author(s)

I very much enjoyed reviewing this paper which addresses a question of interest and importance, about the relations between language and broader social interactive profiles in autistic people. I have only minor comments on the paper, which is well written and comprehensive. I was especially pleased by the admirably respectful language to describe what is known about autism and autistic people.

My comments are as follows:

1. as the authors are well aware, the range of psychological assessments and real world behaviours encompassed within the term "theory of mind" is not consistently agreed upon, and can probably never be resolved. This paper clearly contributes new information to this debate. Nonetheless, I think in the introduction (page 4) it would be possible to be slightly more precise in recognising that "documented differences in the performance of autistic groups on "theory of mind" tasks..." most accurately refers to a fairly narrow definition of ToM focused on automatic inference of complex mental states. For tasks often included within the ToM umbrella - e.g. emotion recognition, empathy - evidence is much less robust (and indeed, readers will discover later that this paper adds pragmatic language to that list). This slightly contrasts with the subsequent statement that, according to Relevance Theory "...theory of mind" is a broad domain incorporating pragmatics..." I wonder if the authors could add a little clarity here regarding the domains linked to ToM that show robust autistic / non-autistic differences and the domains where findings are more unstable, and situate pragmatics - and specifically pragmatic understanding - within this. This would help us understand the predictions about pragmatics in autism in relation to a massive and complex ToM literature, which I appreciate cannot and should not be comprehensively reviewed here. The change might also have some implications for the discussion on page 29.
2. page 7 - the female-weighted sample may not be as unusual as the authors state. I have found similar female-bias especially in online studies with autistic people, though I'm not aware of a citation to support this more generally. it might be more accurate to say that the gender balance of the sample was not representative of the usual gender distribution in diagnosis.
3. page 7: can you briefly mention why some autistic participants did not complete an ADOS? I assume for simple logistical reasons but it would be useful to help judge whether these data are missing at random or not.
4. Table 3 could you include an explicit note to explain what the direction of the effects reported are? i.e. a negative effect size always means lower mean scores in the autistic group, and do those lower scores also then always correspond to "poorer" performance?
5. sex and gender are used interchangeably - most obviously on page 23. Given that this was self-reported I would have thought "gender" would be the most appropriate term to use throughout.
6. page 27, line 32: "...the test battery was devised to put special demand on our ability..." - I suggest replacing "our" with "participants" for greater clarity
7. The discussion spanning pages 32 and 33 neglects to reflect on the fact that communication samples within the ADOS are elicited via the application of specific "presses" designed to exert social and / or communication pressure on the autistic participant. Examiners create a specific kind of communication environment which is calibrated to expose distinctive features of autism in order to make an accurate diagnosis. This is a key factor when attempting to interpret differences between ADOS and self-report communication scores, especially if the authors want to make inferences about the day to day experience of communication for autistic people. On the whole, given the multiple differences between the ADOS and CC-SR measurement contexts, I would cut back on the discussion here which probably relies too much on conjecture.

Review form: Reviewer 2 (Rory Allen)

Is the manuscript scientifically sound in its present form?

Yes

Are the interpretations and conclusions justified by the results?

Yes

Is the language acceptable?

Yes

Do you have any ethical concerns with this paper?

No

Have you any concerns about statistical analyses in this paper?

No

Recommendation?

Accept with minor revision (please list in comments)

Comments to the Author(s)

I note that the authors comment that their results do not support predictions from the Theory of Mind approach when participants made judgements on communicative stimuli. I would suggest the authors consider whether it would be relevant to include references to papers from Ami Klin and Fred Volkmar from 1992 and 1993 which were critical of the Theory of Mind explanation, as well as mentioning the Enactive Mind hypothesis set out by Klin, Volkmar and others in a 2003 Royal Society paper (The Enactive Mind, or from Actions to Cognition: Lessons from Autism). This offered a more developmental approach, whereas the Theory of Mind account is static; while it may be adequate as a broad description of some autistic difficulties, and served an important historical function in changing the way in which we viewed autism, the feeling has been growing that it fails to explain why theory of mind difficulties arise in the first place.

In addition, I think it would be useful for the authors to suggest in their discussion of these results, how they would test samples of children with a similar level of abilities seen in the present paper, to find the developmental precursors of the deficits seen here.

The authors might also consider whether the work of Geoff Bird and co-workers on alexithymia is relevant. Bird has found that alexithymia, which is often co-morbid in autism, explains some of the deficits traditionally explained by autism alone.

Meanwhile this paper is a very welcome and significant contribution to the field, and reinforces the idea that a developmental approach is far more likely to result in properly targeted interventions in autism, than modular accounts based on a one size fits all Theory of Mind model.

Decision letter (RSOS-200845.R0)

Dear Mr Wilson

The Editors assigned to your paper RSOS-200845 "Judging meaning: A domain-level difference between autistic and non-autistic adults" have now received comments from reviewers and would like you to revise the paper in accordance with the reviewer comments and any comments from the Editors. Please note this decision does not guarantee eventual acceptance.

Please submit your revised manuscript and required files (see below) no later than 21 days from today's (ie 27-Aug-2020) date. Note: the ScholarOne system will 'lock' if submission of the revision is attempted 21 or more days after the deadline. If you do not think you will be able to meet this deadline please contact the editorial office immediately.

on behalf of Dr Giorgia Silani (Associate Editor) and Essi Viding (Subject Editor)
openscience@royalsociety.org

Associate Editor Comments to Author (Dr Giorgia Silani):

Comments to the Author:

We have now received the reviews of your manuscript referenced above. The reviewers are generally very positive about your work and clearly in favor of publication. However, they have also listed a number of suggestions which mainly include conceptual/theoretical additions.

These suggestions are outlined in their reviews which have been included below. We invite you to revise the manuscript accordingly.

Reviewer comments to Author:

Reviewer: 1

Comments to the Author(s)

I very much enjoyed reviewing this paper which addresses a question of interest and importance, about the relations between language and broader social interactive profiles in autistic people. I have only minor comments on the paper, which is well written and comprehensive. I was

especially pleased by the admirably respectful language to describe what is known about autism and autistic people.

My comments are as follows:

1. as the authors are well aware, the range of psychological assessments and real world behaviours encompassed within the term "theory of mind" is not consistently agreed upon, and can probably never be resolved. This paper clearly contributes new information to this debate. Nonetheless, I think in the introduction (page 4) it would be possible to be slightly more precise in recognising that "documented differences in the performance of autistic groups on "theory of mind" tasks..." most accurately refers to a fairly narrow definition of ToM focused on automatic inference of complex mental states. For tasks often included within the ToM umbrella - e.g. emotion recognition, empathy - evidence is much less robust (and indeed, readers will discover later that this paper adds pragmatic language to that list). This slightly contrasts with the subsequent statement that, according to Relevance Theory "...theory of mind" is a broad domain incorporating pragmatics..." I wonder if the authors could add a little clarity here regarding the domains linked to ToM that show robust autistic / non-autistic differences and the domains where findings are more unstable, and situate pragmatics - and specifically pragmatic understanding - within this. This would help us understand the predictions about pragmatics in autism in relation to a massive and complex ToM literature, which I appreciate cannot and should not be comprehensively reviewed here. The change might also have some implications for the discussion on page 29.

2. page 7 - the female-weighted sample may not be as unusual as the authors state. I have found similar female-bias especially in online studies with autistic people, though I'm not aware of a citation to support this more generally. it might be more accurate to say that the gender balance of the sample was not representative of the usual gender distribution in diagnosis.

3. page 7: can you briefly mention why some autistic participants did not complete an ADOS? I assume for simple logistical reasons but it would be useful to help judge whether these data are missing at random or not.

4. Table 3 could you include an explicit note to explain what the direction of the effects reported are? i.e. a negative effect size always means lower mean scores in the autistic group, and do those lower scores also then always correspond to "poorer" performance?

5. sex and gender are used interchangeably - most obviously on page 23. Given that this was self-reported I would have thought "gender" would be the most appropriate term to use throughout.

6. page 27, line 32: "...the test battery was devised to put special demand on our ability..." - I suggest replacing "our" with "participants" for greater clarity

7. The discussion spanning pages 32 and 33 neglects to reflect on the fact that communication samples within the ADOS are elicited via the application of specific "presses" designed to exert social and / or communication pressure on the autistic participant. Examiners create a specific kind of communication environment which is calibrated to expose distinctive features of autism in order to make an accurate diagnosis. This is a key factor when attempting to interpret differences between ADOS and self-report communication scores, especially if the authors want to make inferences about the day to day experience of communication for autistic people. On the whole, given the multiple differences between the ADOS and CC-SR measurement contexts, I would cut back on the discussion here which probably relies too much on conjecture.

Reviewer: 2

Comments to the Author(s)

I note that the authors comment that their results do not support predictions from the Theory of Mind approach when participants made judgements on communicative stimuli. I would suggest

the authors consider whether it would be relevant to include references to papers from Ami Klin and Fred Volkmar from 1992 and 1993 which were critical of the Theory of Mind explanation, as well as mentioning the Enactive Mind hypothesis set out by Klin, Volkmar and others in a 2003 Royal Society paper (The Enactive Mind, or from Actions to Cognition: Lessons from Autism).

This offered a more developmental approach, whereas the Theory of Mind account is static; while it may be adequate as a broad description of some autistic difficulties, and served an important historical function in changing the way in which we viewed autism, the feeling has been growing that it fails to explain why theory of mind difficulties arise in the first place.

In addition, I think it would be useful for the authors to suggest in their discussion of these results, how they would test samples of children with a similar level of abilities seen in the present paper, to find the developmental precursors of the deficits seen here.

The authors might also consider whether the work of Geoff Bird and co-workers on alexithymia is relevant. Bird has found that alexithymia, which is often co-morbid in autism, explains some of the deficits traditionally explained by autism alone.

Meanwhile this paper is a very welcome and significant contribution to the field, and reinforces the idea that a developmental approach is far more likely to result in properly targeted interventions in autism, than modular accounts based on a one size fits all Theory of Mind model.

===PREPARING YOUR MANUSCRIPT===

- one version identifying all the changes that have been made (for instance, in coloured highlight, in bold text, or tracked changes);
- a 'clean' version of the new manuscript that incorporates the changes made, but does not highlight them.

This version will be used for typesetting if your manuscript is accepted.

===PREPARING YOUR REVISION IN SCHOLARONE===

Author's Response to Decision Letter for (RSOS-200845.R0)

See Appendix A.

RSOS-200845.R1 (Revision)

Review form: Reviewer 1 (Sue Fletcher-Watson)

Is the manuscript scientifically sound in its present form?

Yes

Are the interpretations and conclusions justified by the results?

Yes

Is the language acceptable?

Yes

Do you have any ethical concerns with this paper?

Yes

Have you any concerns about statistical analyses in this paper?

No

Recommendation?

Accept as is

Comments to the Author(s)

Thank you for this thoughtful response to my previous comments. I think the manuscript is now in very strong shape: it makes an articulate, and well-supported argument and shares original findings with clarity.

Review form: Reviewer 2 (Rory Allen)

Is the manuscript scientifically sound in its present form?

Yes

Are the interpretations and conclusions justified by the results?

Yes

Is the language acceptable?

Yes

Do you have any ethical concerns with this paper?

No

Have you any concerns about statistical analyses in this paper?

No

Recommendation?

Accept as is

Comments to the Author(s)

Thank you. I think the paper represents a valuable contribution to the literature and I have no further suggestions to make.

Decision letter (RSOS-200845.R1)

Dear Mr Wilson,

It is a pleasure to accept your manuscript entitled "Judging meaning: A domain-level difference between autistic and non-autistic adults" in its current form for publication in Royal Society Open Science. The comments of the reviewer(s) who reviewed your manuscript are included at the foot of this letter.

===COVID-SPECIFIC TEXT -- WILL ONLY BE ADDED TO COVID-PAPERS BY THE EDITORIAL OFFICE===

COVID-19 rapid publication process:

We are taking steps to expedite the publication of research relevant to the pandemic. If you wish, you can opt to have your paper published as soon as it is ready, rather than waiting for it to be published the scheduled Wednesday.

This means your paper will not be included in the weekly media round-up which the Society sends to journalists ahead of publication. However, it will still appear in the COVID-19 Publishing Collection which journalists will be directed to each week (<https://royalsocietypublishing.org/topic/special-collections/novel-coronavirus-outbreak>).

If you wish to have your paper considered for immediate publication, or to discuss further, please notify openscience_proofs@royalsociety.org and press@royalsociety.org when you respond to this email.

===END OF COVID-SPECIFIC TEXT -- WILL BE REMOVED AS NECESSARY BY THE EDITORIAL OFFICE===

on behalf of Dr Giorgia Silani (Associate Editor) and Essi Viding (Subject Editor)
openscience@royalsociety.org

Reviewer comments to Author:
Reviewer: 2

Comments to the Author(s)

Thank you. I think the paper represents a valuable contribution to the literature and I have no further suggestions to make.

Reviewer: 1

Comments to the Author(s)

Thank you for this thoughtful response to my previous comments. I think the manuscript is now in very strong shape: it makes an articulate, and well-supported argument and shares original findings with clarity.

Appendix A

Dear Dr Silani,

Many thanks for sending us through comments from the reviewers. We are very appreciative of the positive feedback. Below we reproduce the peer reviews, and offer responses to the suggested edits and conceptual additions.

Associate Editor Comments to Author(s) (Dr Giorgia Silani):

We have now received the reviews of your manuscript referenced above. The reviewers are generally very positive about your work and clearly in favor of publication. However, they have also listed a number of suggestions which mainly include conceptual/theoretical additions. These suggestions are outlined in their reviews which have been included below. We invite you to revise the manuscript accordingly.

Reviewer 1 Comments to Author(s):

I very much enjoyed reviewing this paper which addresses a question of interest and importance, about the relations between language and broader social interactive profiles in autistic people. I have only minor comments on the paper, which is well written and comprehensive. I was especially pleased by the admirably respectful language to describe what is known about autism and autistic people.

My comments are as follows:

1. as the authors are well aware, the range of psychological assessments and real world behaviours encompassed within the term "theory of mind" is not consistently agreed upon, and can probably never be resolved. This paper clearly contributes new information to this debate. Nonetheless, I think in the introduction (page 4) it would be possible to be slightly more precise in recognising that "documented differences in the performance of autistic groups on "theory of mind" tasks..." most accurately refers to a fairly narrow definition of ToM focused on automatic inference of complex mental states. For tasks often included within the ToM umbrella - e.g. emotion recognition, empathy - evidence is much less robust (and indeed, readers will discover later that this paper adds pragmatic language to that list). This slightly contrasts with the subsequent statement that, according to Relevance Theory "...theory of mind" is a broad domain incorporating pragmatics..." I wonder if the authors could add a little clarity here regarding the domains linked to ToM that show robust autistic / non-autistic differences and the domains where findings are more unstable, and situate pragmatics - and specifically pragmatic understanding - within this. This would help us understand the predictions about pragmatics in autism in relation to a massive and complex ToM literature, which I appreciate cannot and should not be comprehensively reviewed here. The change might also have some implications for the discussion on page 29.

Thank you for raising this important point. We have outlined our take on “theory of mind” and some of the conceptual issues around the construct more fully in the introduction. We have also made some small amendments to the discussion based on the comments of yourself and the other reviewer regarding “theory of mind”.

2. page 7 - the female-weighted sample may not be as unusual as the authors state. I have found similar female-bias especially in online studies with autistic people, though I'm not

aware of a citation to support this more generally. it might be more accurate to say that the gender balance of the sample was not representative of the usual gender distribution in diagnosis.

This is a good point. We have also noted that other researchers sometimes report a similar gender distribution in their samples, and so we have amended this comment in line with your suggestion.

3. page 7: can you briefly mention why some autistic participants did not complete an ADOS? I assume for simple logistical reasons but it would be useful to help judge whether these data are missing at random or not.

Six of 71 participants did not complete an ADOS assessment. This was generally because of logistical reasons, e.g. the participant lived too far away or there were scheduling issues. We have added this information to the methods.

4. Table 3 could you include an explicit note to explain what the direction of the effects reported are? i.e. a negative effect size always means lower mean scores in the autistic group, and do those lower scores also then always correspond to "poorer" performance?

Negative values always indicated lower average scores. We have clarified this in the table legend.

5. sex and gender are used interchangeably - most obviously on page 23. Given that this was self-reported I would have thought "gender" would be the most appropriate term to use throughout.

Generally, we tried to use “gender” for the reason you suggest, although there were a couple of occasions on which we slipped up on this. “Sex” has been amended to “gender” throughout.

6. page 27, line 32: "...the test battery was devised to put special demand on our ability..." - I suggest replacing "our" with "participants" for greater clarity

We agree this is ambiguously worded, and have edited it to “put special ability on the ability...”.

7. The discussion spanning pages 32 and 33 neglects to reflect on the fact that communication samples within the ADOS are elicited via the application of specific "presses" designed to exert social and / or communication pressure on the autistic participant. Examiners create a specific kind of communication environment which is calibrated to expose distinctive features of autism in order to make an accurate diagnosis. This is a key factor when attempting to interpret differences between ADOS and self-report communication scores, especially if the authors want to make inferences about the day to day experience of communication for autistic people. On the whole, given the multiple differences between the ADOS and CC-SR measurement contexts, I would cut back on the

discussion here which probably relies too much on conjecture.

We have briefly included this observation in the discussion, as well as editing the section down for greater concision as we agree it was quite speculative.

Reviewer 2 Comments to Author(s):

I note that the authors comment that their results do not support predictions from the Theory of Mind approach when participants made judgements on communicative stimuli. I would suggest the authors consider whether it would be relevant to include references to papers from Ami Klin and Fred Volkmar from 1992 and 1993 which were critical of the Theory of Mind explanation, as well as mentioning the Enactive Mind hypothesis set out by Klin, Volkmar and others in a 2003 Royal Society paper (The Enactive Mind, or from Actions to Cognition: Lessons from Autism). This offered a more developmental approach, whereas the Theory of Mind account is static; while it may be adequate as a broad description of some autistic difficulties, and served an important historical function in changing the way in which we viewed autism, the feeling has been growing that it fails to explain why theory of mind difficulties arise in the first place.

Thank you for your thoughts about the limitations of the “theory of mind” account. We have previously found the Klin et al. (2003) paper very useful, and agree that it is a worthy addition to the discussion. Also, we have made a few edits to the introduction and discussion around the “theory of mind” construct in line with comments from yourself and the other reviewer.

In addition, I think it would be useful for the authors to suggest in their discussion of these results, how they would test samples of children with a similar level of abilities seen in the present paper, to find the developmental precursors of the deficits seen here.

In the discussion, we briefly mention that future longitudinal research could track developmental change in language, communication and “theory of mind” on similar tests administered to child samples.

The authors might also consider whether the work of Geoff Bird and co-workers on alexithymia is relevant. Bird has found that alexithymia, which is often co-morbid in autism, explains some of the deficits traditionally explained by autism alone.

This is a valid point, and we agree that there is certainly some overlap between “theory of mind” and alexithymia, e.g. performance on the “Reading the Mind in the Eyes” test, traditionally a measure of “theory of mind” though with an obvious demand on emotion recognition, is closely related to alexithymia (Oakley et al., 2016). We are less sure how alexithymia would relate to our tests, so we have avoided speculating on this in the discussion, and unfortunately, we cannot test this empirically as we did not administer an alexithymia measure. This would be interesting to explore in future research.

Meanwhile this paper is a very welcome and significant contribution to the field, and reinforces the idea that a developmental approach is far more likely to result in properly

targeted interventions in autism, than modular accounts based on a one size fits all Theory of Mind model.